# Effect of the ActivaMotricidad Program on Improvements in Executive Functions and Interpersonal Relationships in Early Childhood Education

**DOI:** 10.3390/jfmk9040231

**Published:** 2024-11-12

**Authors:** Nuria Ureña Ortín, Iker Madinabeitia Cabrera, Francisco Alarcón López

**Affiliations:** 1Department of Plastic, Musical, and Dynamic Expression, University of Murcia, 30100 Murcia, Spain; nuriaur@um.es; 2Department of General Didactics and Specific Didactics, University of Alicante, Carretera San Vicente del Raspeig, S/N, 03690 San Vicente del Raspeig, Spain; f.alarcon@ua.es

**Keywords:** self-regulation, interpersonal skills, physical education, preschool, comprehensive program, active breaks, mindfulness

## Abstract

**Background**: The objective of this study was to analyze the impacts of a comprehensive physical exercise program with cognitive involvement during the school day on the executive functions and interpersonal skills of 5- and 6-year-old children. **Methods**: A total of 68 children participated in a 3-week pre–post intervention. **Results**: The results showed significant improvements in executive functions and interpersonal skills under the experimental conditions compared to the control group. These positive effects were observed to persist after a three-week follow-up. **Conclusions**: Therefore, the ActivaMotricidad program, which focuses on developing fundamental motor skills through cognitive, cooperative, and coordinative challenges, can serve as a useful and low-cost tool to achieve the objectives of comprehensive development and sustainability for children in early childhood education.

## 1. Introduction

The emergence of cognitive control during childhood fosters greater autonomy and increasingly adaptive behavior [1], which translates into better behavioral self-regulation [2]. This development intensifies between the ages of 3 and 6 and remains relevant throughout life, influencing social, psychological, and cognitive well-being [3]. Furthermore, a correlation between executive functions (EFs) and self-regulation capacity has been demonstrated [4,5]. The literature suggests that behavioral self-regulation is a manifestation of EFs, and its assessment can provide an indirect measure of these functions [6,7]. Additionally, higher self-regulation and EFs in preschool children facilitate cognitive reasoning in school [8] and better performance in daily school tasks [9,10,11], motivating researchers to seek strategies for developing these cognitive capacities. In the context of self-regulation and EFs, social behavior is crucial during childhood, facilitating the formation of social bonds and moderating aggression [12]. However, it is important to note that this relationship does not necessarily imply direct causality. Various confounding factors, such as socioeconomic status, family environment, and genetic influences, can impact both the development of executive functions and self-regulation in children [13,14].

Previous studies have demonstrated the effects of cooperative games on social, emotional, and psychomotor development variables [15]. Recently, it was found that children’s participation in playful and active interventions contributed to improving their social relationships [16]. Similarly, it was observed that the inclusion of play within the classroom in early childhood education had a positive effect on interpersonal skills after a four-month period [17]. However, the degree of competitiveness [16] and complexity [17] may not promote the same levels of cooperation and interpersonal skills. Therefore, it is essential to consider the specific characteristics of play when assessing its impact on child development. For example, the development of comprehensive methods that consider physical, cognitive, and social aspects has shown greater effectiveness in behavioral self-regulation and interpersonal skills than those that have taken an analytical approach, such as isolated cognitive training [13,14]. For instance, a meta-analysis found that training EF skills can be more effective for children when integrated into daily activities, such as games with constant challenges [18].

Early childhood education centers are crucial for cognitive and social development as children spend many hours a day in them. At the same time, integrating intervention programs within the school environment that use play and motor activity finds benefits in EFs and behavioral self-regulation [4,5]. Similarly, motor games in which participants must work together to achieve common goals have been shown to improve social skills and peer collaboration [19]. This approach is consistent with the framework that explains the human need to play [20]. The authors suggest that new computational models like “Predictive Processing” could explain cognition and action, and their relationships through a single mechanism called prediction error minimization, where the brain tries to reduce the mismatch between how it perceives the world and the prediction of how it might be [21]. The model suggests that children deliberately seek surprising situations to resolve prediction error, resulting in a positive sensation. This model not only unifies various perspectives on play and learning but also integrates the role of play in enhancing cognitive skills [20].

However, it is not clear that increasing physical exercise in early childhood has a facilitating effect on the development of EFs and interpersonal skills. Interventions on the potential “brain, cognitive, and academic” benefits in children aged 5 to 13 have found low to moderate effects [22]. Moreover, these effects vary according to the specific characteristics of the interventions [23]. When comparing the type of intervention, those that include social and/or cognitive demands seem to be more effective than those that focus more narrowly on increasing physical activity [24,25]. Consequently, the study of the qualitative characteristics of practice has gained special interest. This has led researchers to analyze the impact of cognitively challenging tasks. When interventions using physical exercise that include some form of cognitive engagement have been studied, compared to a control condition, better cognitive performance is found in the experimental group [26].

Similarly, interventions with dual tasks simultaneously (motor and cognitive), through games enriched with cognitive challenges, can improve declarative memory [27]. This combined practice is more cognitively demanding, as it requires integrating and coordinating the two tasks simultaneously, which involves additional cognitive processing [28]. The environmental conditions of the games used in our intervention proposal force children into a continuous process of control and flexibility to adjust their actions to the changing demands of the tasks, which is believed to strengthen the component processes of EFs [29,30]. In this scenario, interventions through cooperative motor activities help children develop social skills, self-control, and foster interpersonal relationships with their peers [31].

On the other hand, the development of fundamental motor skills, defined as the basic skills necessary to participate in a variety of physical activities, including locomotor, object control, and stability skills, are determinants for the cognitive development of preschool children [32]. Longitudinal studies with preschoolers provide evidence that they are predictive of working memory updating [33] and overall EF performance [34]. Additionally, programs that include fundamental motor skills training have been found to benefit cognitive development (including EFs) and academic learning in children aged 3 to 7 years [35].

From the neuroscientific perspective of movement, the cognitive efforts required to develop new or complex motor skills could explain the improvements in EFs, as they affect the central processes that implement cognitive control. It is known that while performing a difficult motor task, cognitive control is needed to adjust behavior according to its goals [36], leading to more consistent neuronal plasticity changes [37].

While some previous studies have investigated these interventions in different populations, the underlying principles and physiological and cognitive mechanisms are applicable and relevant to our target population of preschool children. In our study, physical, cognitive, and interpersonal activities were adapted and personalized to address the needs and characteristics of preschool children, following the recommendations of the existing literature [32]. This was achieved by adapting the level of difficulty and type of tasks to maintain an adequate and continuous challenge [38]. However, to date, the combined impact of physical and socio-cognitive activities in a structured intervention context in early childhood has not been evaluated. Our integrative approach seeks to fill this gap by systematically combining these elements. Thus, the objective of this research was to analyze the impact of increasing physical education with cognitive and social involvement during school hours, through a comprehensive program called ActivaMotricidad, on the EFs and interpersonal skills of 5- and 6-year-old children. This program includes fundamental motor skills, cognitively challenging cooperative motor games and stories, specifically designed to increase both physical activity and children’s cognitive and interpersonal participation [32,39]. We hypothesized that the experimental group, subjected to the ActivaMotricidad program, will show better EFs and greater interpersonal relationship capacity at the end of the intervention compared to the control group participants.

## 2. Materials and Methods

### 2.1. Participants

The participants included an initial sample of 70 children aged 5 to 6 years distributed in three classes (mean = 5.46 years; SD = 0.22) enrolled in a public educational center offering levels from preschool to primary education. Two children were excluded from the study for receiving support for specific educational needs. The final sample consisted of 68 participants. Parents were informed about the study and given the option for their children not to participate in data collection. All parents provided informed consent for their children’s participation. The research was conducted in accordance with the Declaration of Helsinki (2013) on human research, and the protocol was approved by the Ethics Committee of the University of Murcia (D: 2884/2020). The following exclusion criteria were applied: (a) missing more than three recess and/or active break sessions, (b) not completing the three motor story sessions, (c) having a prior history of pathologies, and (d) not providing informed consent.

### 2.2. Design

This study followed a cluster randomized controlled trial (cRCT) design at the class level with pretest–posttest and a retention test for the experimental group. Participants (in this case, the three classes) were randomly assigned to intervention (2 classes) and control groups (1 class). This cRCT design is widely used in research to robustly evaluate the effect of interventions, as demonstrated in studies where groups were randomly assigned to classes [40,41]. The independent variable was the type of intervention, while the dependent variables were performance on the HTKS behavioral test and interpersonal relationships. The children were unaware of the study’s purpose and the experimental conditions. Teachers were unaware of the conditions and were informed that they would receive access to the program resources once the study was completed. To verify whether the effects were maintained over time, a retention retest was conducted for the experimental group. Similarly, once the trial was completed, the control group received a delayed implementation of the program to benefit from its potential positive effects, which is why the control group did not have a retention measure.

### 2.3. Procedure

The intervention was implemented at the class level by two researchers during regular classroom activities and school hours. The study was conducted in three preschool classrooms, corresponding to groups A, B, and C, distributed according to the school year (see Table 1). Randomization was performed to assign the experimental conditions and establish the order. Classrooms A and B, with a total of 47 children, were the experimental group, which, after three weeks, followed the control intervention to analyze retention. Meanwhile, classroom C, with 21 children, was the control group. This class distribution was due to the limited access to three classes and potential challenges, including experimental mortality and non-compliance with conditions, so two classes were assigned to the experimental group to allow flexibility in the intervention.

Outcome variables were measured for all classes in a pretest and final test after 3 weeks. The evaluation was conducted by a group of trained and blinded researchers. EFs and interpersonal skills were assessed. The intervention was first carried out in the experimental group, while the control group maintained its usual routine with one 50 min physical education session per week. The control group teacher was instructed to continue with their usual practices without any changes in their usual instruction format.

### 2.4. Intervention

The ActivaMotricidad program is a comprehensive approach that combines the following three main features: the motor dimension through fundamental motor skills, cognitive challenges [42], and interpersonal problem-solving situations [43] using active, globalizing, and interdisciplinary methodologies typical of early childhood education. The researchers implemented the program over 3 weeks in classrooms A–B and in classroom C during the following 3 weeks, in winter and before the Easter holidays.

The ActivaMotricidad program guide included 15 active breaks (ABs), each lasting 10–15 min. This resource is characterized by incorporating classroom breaks with medium-high intensity and short-duration physical exercises. These physical exercise periods were conducted at 10:15 a.m. using two methodological strategies. On the one hand, motor games were employed, based on the motor dimension and introducing self-regulation variables, such as attention, response flexibility, and the inhibition of one motor action in favor of a more effective one. Cooperative problem-solving strategies were also articulated. Additionally, once a week, 3 ABs with a mindfulness focus, based on Snel’s proposal, were included as a means to promote children’s attention [44,45]. The program also integrated three motor stories (MSs), each lasting 60 min, organized in various moments or phases [46] and with a frequency of once a week. The narration of the MS fostered character identification, working memory, and role exchange. It also promoted the coordination and adaptation of each child’s motor actions with others, accepting and valuing the proposals of other peers to enhance the acquisition of social skills and cognitive strategies for interpersonal problem-solving. Finally, for unstructured active recess, 15 sessions with 8 learning environments (LEs) were conducted through painted marks on the ground, manipulating both space and materials, lasting 30 min and with a frequency of 5 times per week. Unstructured play allowed children a space to autonomously choose and create their own playful activities, navigate their social worlds, make independent decisions, and take responsibility for deciding how to navigate the movement environment and experience the consequences of their own actions [47]. To clarify the type of motor activities contained in the program, Figure 1 presents examples of them, along with a link to the complete program, in which is described how tasks were designed with varying incremental levels of difficulty, incorporating additional elements to be remembered and/or manipulated in short-term memory (i.e., updating). It also details the rules used to introduce new regulations that require resolving changes in relation to previous rules or dominant behaviors (i.e., inhibition). Additionally, all activities provided instructions to increase the challenge of the activity as children became more competent [48]. This is illustrated in Table 2, in the “Cognitive Involvement” section.

The construct validity was qualitatively evaluated. Two psychology experts and three experts in physical activity and sports sciences rated the adequacy of the program’s motor activities to stimulate EFs and interpersonal skills. Discrepancies were resolved through expert discussion [49]. The ActivaMotricidad program (Table 2) was designed to be compatible in a variety of early learning contexts and is available in the book *ActivaMotricidad* [46].

**Table 2 jfmk-09-00231-t002:** Summary of the ActivaMotricidad program’s contents.

PROGRAM	ActivaMotricidad: Physical Exercise, Cognitive Involvement, and Interpersonal Relationships			
Methodological Resources	Active breaks through motor games and mindfulness practices, motor stories with cooperative narratives, and learning environments with free play			
Modality	AB	AB	Specific Physical Education Session	Breaks
Space	Classroom and/or Nearby Outdoor Patio	Classroom	Sports Pavilion	Outdoor Patio
Number	12 ABs Motor Games	3 ABs Mindfulness	3 MSs Cooperative Activities	8 LEs Free Motor Play
Time (minutes)	10–15 min	60 min	30 min	10–15 min
Duration	3 Weeks			
Frequency (days/week)	4 Monday, Tuesday, Wednesday, and Thursday	1 Friday	1 Tuesday	5 Monday to Friday
School Day Timing	10:15 to 10:30	11:30 to 11:45	12:30 to 13:30	11:00 to 11:30
Cognitive Involvement	To increase the likelihood of reaching the children’s optimal challenge point, each task was designed with different incremental levels of difficulty [48,50], as follows: (a) adding more elements to be remembered and/or manipulated in short-term storage (i.e., updating); (b) introducing new rules requiring resolution of the change with the previous rule or a dominant behavior (i.e., inhibition); and (c) including several sets of rules to switch fluidly (i.e., shifting).			
Interpersonal Relationships	The following are basic elements of [51]: (1) highlighting the importance of cooperative skills; (2) ensuring children understood the cooperative skills of each motor proposal; (3) organizing cooperative situations and providing opportunities for children to practice repeatedly throughout the program; (4) organizing group activities and offering opportunities for introspection, as group activities were designed to provide opportunities for active feedback from children; (5) encouraging students to practice cooperative skills continuously until full internalization: full internalization is expected when cooperative skills are practiced repeatedly in learning situations.			

ABs: active breaks; MSs: motor stories; LEs: learning environments.

### 2.5. Verification of the Fidelity of the Program Implementation

To verify the quality of the program implementation, the following measures were taken. First, the program was carried out by two researchers with initial training in Education Sciences and at least two years of work experience in early childhood education. This training ensures that the program is executed through a high-quality teaching intervention.

Additionally, the researchers received prior training to ensure the correct application of the methodology designed for the intervention [48]. This training covered the “core components” of the program, the temporal distribution of each of these core components, and how to use the assessment instruments for both dependent variables and those related to checking the success of the experimental manipulations.

On the other hand, a “manipulation check” was conducted. To verify that the application of the ActivaMotricidad program had an optimal level of difficulty to generate adequate cognitive engagement in the participants, the adaptation of the Borg scale (RPE) by authors such as Egger et al. [52] was used. At the end of each week, participants answered the question (PCG), “How much did you have to concentrate (think) to do the activity?”, using a Likert scale from 1 to 3, where 1 corresponded to “little” and 3 to “a lot” [53]. The researchers used the recorded PCG values to adjust the difficulty level of the motor tasks for the following week. A verification and control of motor participation time in unstructured active recesses was also carried out through a log sheet. It ensured motor involvement and the time children were participating in movement in the learning environments.

### 2.6. Instruments

**Behavioral Self-Regulation (Executive Functions):** The Spanish version of the Head-Toes-Knees-Shoulders (HTKS) task was used to assess behavioral self-regulation [54]. The extended version of this task is a behavioral measure that requires updating working memory, motor inhibition, and response flexibility [2,55]. The HTKS has been used with a large number of early childhood populations and has good construct validity and reliability [55,56]. Previous studies have found that the task is related to all aspects of EFs [57]. To complete the HTKS, children had to do the opposite of what the researcher said. For example, when children were asked to “touch their toes”, they had to touch their head and vice versa. Therefore, children not only had to retain the task rules in memory but also inhibit the tendency to give a natural response (e.g., avoid touching their toes when told “touch your toes”) while responding oppositely (e.g., touching their head). The task also involved increasing cognitive demands as the task rules became more complex, adding a new set of commands (“touch your shoulders” and “touch your knees”).

The HTKS is divided into two parts, taking approximately 3–5 min to complete. Part I consists of 5 familiarization trials and 10 test trials, during which children responded to one of two paired commands (“touch your head” or “touch your toes”) with the opposite response. Each incorrect response was coded as 0 points, a self-corrected response was coded as 1 point, and a correct response was coded as 2 points. Self-correction is defined as any movement directed toward the incorrect response but ending with the correct action. In Part II, two additional commands (“touch your shoulders” and “touch your knees”) were added. After completing 5 practice trials with only the new commands, children completed 10 test trials in which they had to respond to the four previously mentioned commands. In total, there are 30 trials in the entire task with a possible score range of 0 to 60. Before conducting the assessment using the HTKS, the researchers were trained through a pilot study to improve their skills in using the instrument.

**Interpersonal Problem-Solving Test (TREPI):** The task involved presenting participants with three problematic situations that may arise in social interaction, depicted through a drawing and a short story, which were close to the child’s life. For example, the first situation referred to “a girl who did not want to let her younger brother play with her doll but was afraid her mother would scold her for it”. The child observed each of the three cards representing different situations evidencing interpersonal conflicts and had to answer the following question for each situation: “What do you think is the best thing this boy or girl can do to solve their problem?” One point was awarded for each prosocial response to resolving the problem situation, providing correction and interpretation patterns for the TREPI. The responses children gave to the three problematic social situations presented in the test were classified into the following two categories: (a) positive or socially favorable responses and (b) negative or socially disruptive responses. The test was administered individually, in a room where the researcher and the child were present, and lasted 15 min per child [43]. Before conducting the assessment, the researcher was trained through a pilot study to improve their skills in using the instrument. This test is commonly used for preschool children (4–6 years) with psychometric guarantees of reliability and validity [15].

**Active Recess Observation System in Early Childhood (SOREACI):** To evaluate the motor engagement time of children during unstructured active recess, an observational recording system was designed, following the methodology of Thomas and Nelson [58]. This system classified children into the following two categories: physically active and inactive. A child was considered inactive if they remained seated and/or standing without moving for 5 consecutive minutes or more. Conversely, they were classified as active if the child was walking, moving, running, and jumping, among other motor activities. Prior to implementing the system, the researcher received practical training to ensure the accuracy and consistency of the observations. This instrument allowed for direct and systematic in situ observation, that is, in the same environment where the recess took place, with the aim of documenting children’s movement during free play. During the three weeks of intervention, the motor activity of the children was recorded and classified for each recess. Children who were punished or finishing their lunch were excluded from observation.

### 2.7. Statistical Analysis

Data from all considered variables in the sample were grouped. Despite being aware that the sample may be small and uneven, a non-parametric statistical analysis was chosen to address this issue, as it is recognized that non-parametric analysis can obtain robust and reliable results despite sample size limitations. In any case, the Shapiro–Wilk normality test was applied to each variable to initially determine the choice between parametric or non-parametric analysis as appropriate. Initially, an analysis was employed to verify that the time dedicated to physical exercise was homogeneous in the sample, based on the total minutes of exercise. Then, to evaluate the differences between pre- and post-measurements in, on the one hand, the experimental group and, on the other hand, the control group in the HTKS and TREPI variables, a paired samples analysis was conducted. Similarly, the same analysis was performed to check whether there was retention of effects in the experimental group between posttest and retest measurements. Finally, to determine whether there were significant differences between the two groups in pre- and post-measurements, an independent samples analysis was conducted.

The significance level was set at 0.05. To measure effect size, Vargha and Delaney’s A was used for all analyses. It is interpreted as follows: between 0.56 and less than 0.64 for small effects and between greater than 0.44 and 0.5 for small effects in the opposite direction; between 0.64 and less than 0.71 for medium effects, and between greater than 0.29 and 0.34 for medium effects in the opposite direction; 0.71 or greater for large effects and 0.29 or less for large effects in the opposite direction. For the statistical analysis, RStudio software (version 2023.06.0) was used.

## 3. Results

Table 3 presents the descriptive data of the study variables. The Shapiro–Wilk test indicated that the data did not meet normality standards, so non-parametric statistics were used for all analyses. Additionally, the Mann–Whitney test revealed that physical activity levels (SOREACI) were homogeneous across the sample, as no significant results were indicated (W = 593; *p* = 0.112). Furthermore, no differences were observed among classes in the pre-measurement.

### 3.1. Effect of Groups on HTKS and TREPI

Regarding the HTKS, the Wilcoxon paired samples analysis showed that the experimental group exhibited significant improvement (V = 71, *p* < 0.001, A = 0.118: large effect), as well as the control group (V = 17, *p* = 0.001, A = 0.292: medium effect). In the case of the TREPI, similarly, the experimental group showed a significant improvement (V = 52, *p* < 0.001, A = 0.274: medium effect), while the control group showed no significant differences. Regarding the retention of effects in the experimental group, a significant result was observed in the HTKS (V = 9, *p* < 0.001, A = 0.352: small effect) but not in the TREPI. However, this result does not indicate a significant increase as the values remain practically unchanged; rather, it shows that the data variability is lower. Part of these results can be seen in Figure 2.

### 3.2. Results According to the Difference in Effect Between Groups

Starting with the HTKS, the Mann–Whitney test showed that the experimental group improved significantly more than the control group (W = 233.5, *p* < 0.001, A = 0.763: large effect). In the case of the TREPI, the same significant result was observed (W = 296, *p* = 0.0081, A = 0.700: medium effect). Again, these results are visually shown in Figure 2.

## 4. Discussion

This study explored the effects on EFs and interpersonal relationship capacity of preschool children who participated in a comprehensive school-based intervention program involving physical exercise with high cognitive and social demands. Both the program and usual practice had a positive effect on EFs. Compared to the control group, improvements in EFs were significantly greater for the experimental group. The changes observed in the experimental group were maintained after 3 weeks of returning to the usual routine, suggesting the effectiveness and sustainability of the ActivaMotricidad program over time. However, because of the lack of a control group during the retention period, additional studies are recommended to confirm these findings. In contrast, for interpersonal skills, improvements were observed exclusively after the ActivaMotricidad program.

The positive effect of the intervention on children’s EFs beyond the rapid development of these skills in preschool years [59] is consistent with the evidence of the need to support and improve children’s EFs [29,48], and to do so through integrative approaches [48,60]. Unlike most existing proposals, which prioritize less applicable contexts, such as individual and/or computerized administration (which commonly require professional administration and resource availability), the ActivaMotricidad program represents a real and low-cost alternative to these approaches that can be applied as a “menu” of practices, activities, and resources to flexibly adapt to different contexts. The accessibility and acceptability of the current approach create a unique opportunity for integrated practices that produce benefits for preschool children, thus confirming the initial hypothesis.

Our previous findings support the current results. For example, the acute effects of a 15 min AB with various levels of difficulty in children aged 4 to 5 years were analyzed [61], finding improvements in EFs regardless of difficulty. These results were also found in other studies [62]. After implementing a comprehensive movement education program for 5-year-old children, which included learning situations similar to ours (group activities, cooperative games, and social interaction), improvements in behavioral self-regulation were also found.

The current results also reinforce those found by previous studies that use both motor skills development and physical fitness improvement in their proposals. Some early childhood intervention studies based on movement or including games that require movement reported improvements in EFs [48]. In contrast, others only found maintenance of initial values, in contrast to the control group, which worsened [4]. This heterogeneity could be explained by the lack of control over task complexity. As Antunes et al. [63] point out, participants need to exceed a minimum stimulation threshold to facilitate cognitive processes and promote changes in EFs. In fact, after reviewing programs that influence EFs, Diamond and Ling [29] established that to identify differences between treatment and control groups, tasks must require participants to push their cognitive skills to their limits. In our study, all tasks were designed to have specific cognitive implementation and adjustment based on the perceived cognitive engagement of the participants.

In addition to the specific interventions of the ActivaMotricidad program, it is crucial to consider the role of learning environments and settings in promoting a mastery climate focused on developing interpersonal skills, personal improvement, and effort. This approach supports students’ self-regulation and intrinsic motivation. Studies have demonstrated that a mastery climate improves academic engagement and EFs [64]. Integrating these strategies into educational interventions maximizes positive effects on children’s skills. Similar to the study by Robinson et al. [4], in the ActivaMotricidad program, active recesses were designed to grant autonomy and responsibility to children, allowing them to decide how to engage in the movement environment. This included choosing learning environments, difficulty levels, time allocation, and grouping for interaction with peers. The results obtained regarding active recess indicated that participants remained active during recess time, using the different motivating activity environments, thus increasing the time dedicated to physical activity during the school day. The results confirmed that active recess was a fundamental moment within the school environment to promote physical activity. These results are consistent with studies that have shown that recess is a good setting to increase physical activity [65,66,67].

Regarding interpersonal skills, the TREPI results indicated that the program has a positive effect. This may be due to the cooperative motor games in the program, which involve tasks requiring sharing, task-solving, or decision making. Garaigordobil and Berrueco [43] found similar benefits following the application of cooperative games. At these ages, motor play has positive repercussions for emotional regulation, conflict management, social interaction, positive prosocial behaviors [68], personal conflict resolution [69], and resilience development [70] while also reducing problematic behaviors in class [71].

The study’s limitations include the sample size and that it is not completely randomized. In any case, a larger sample size and a longer implementation period are needed to generalize and transfer these results to the population. Additionally, although as stated in Table 2, in the “Cognitive Involvement” section, we tried to ensure that the tasks always presented a physical and cognitive challenge, making modifications accordingly, not all participants will gain the same benefits if individual differences in learning styles, cognitive development, or social skills are not addressed. Therefore, future studies should control for initial levels when designing tasks to assess the potential degree of improvement. Moreover, while a cRTC design is a valid approach for controlling external factors such as lifestyle influences, it would be beneficial for future studies to evaluate these factors specifically and include them in the statistical analyses. Furthermore, this study utilized the SOREACI to assess children’s motor engagement during unstructured playtime; however, future research could also consider including specific types of activities chosen by the children, their interactions with peers, and their play styles.

In summary, the development of fundamental motor skills, which are cognitively stimulating, will improve children’s personal and social competences [72]. Therefore, we can affirm the importance of comprehensive programs that use methodologies appropriate for children’s developmental stage in early childhood, being an excellent resource to incorporate into daily routines and improve the mentioned skills.

## 5. Conclusions

The results obtained suggest that quality pedagogical approaches with challenging levels of physical, social, and cognitive exercises integrated into preschool routines are optimal for promoting self-regulation and interpersonal relationships in the early years of life. It is important to avoid generalizations, as the findings are directly related to the specific characteristics of the ActivaMotricidad program and the studied population. Comprehensive physical exercise programs could be the basis through which to achieve cognitive, motor, and social benefits for children. The key to the success of the ActivaMotricidad program lies in its comprehensive approach, which combines physical activities with cognitive challenges designed to simultaneously stimulate children’s motor, cognitive, and social development. This integrated approach leverages the synergy between physical activity and cognitive stimulation, which has been shown to be effective in previous studies and in our own work.

This study was positively received by preschool educators and children, demonstrating the feasibility and applicability of a physical exercise program that harmonizes with the academic teaching of the school day and offers potential improvements in children’s cognitive and social abilities. It is crucial to promote early childhood interventions for adequate school preparation, outstanding academic performance, physical literacy, and future success in professional and personal life. The contribution of this program lies in the innovative integration of age-appropriate methodological resources during the school day, including cognitive and social participation with physical exercise in young children.

## Figures and Tables

**Figure 1 jfmk-09-00231-f001:**
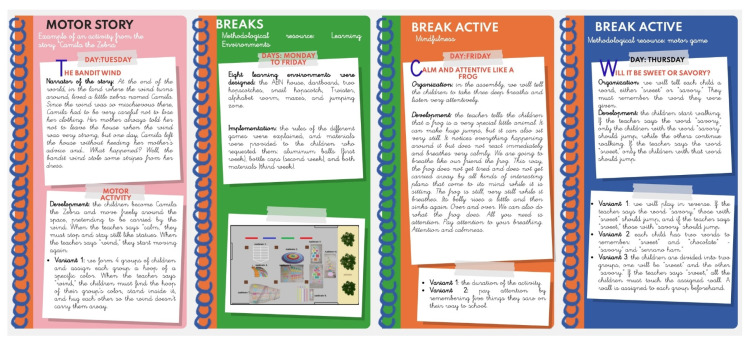
Example of the activities in the ActivaMotricidad program (https://view.genially.com/666b23ae83c14d0014215eef/mobile-ejemplo-activamotricidad, accessed on 9 October 2024).

**Figure 2 jfmk-09-00231-f002:**
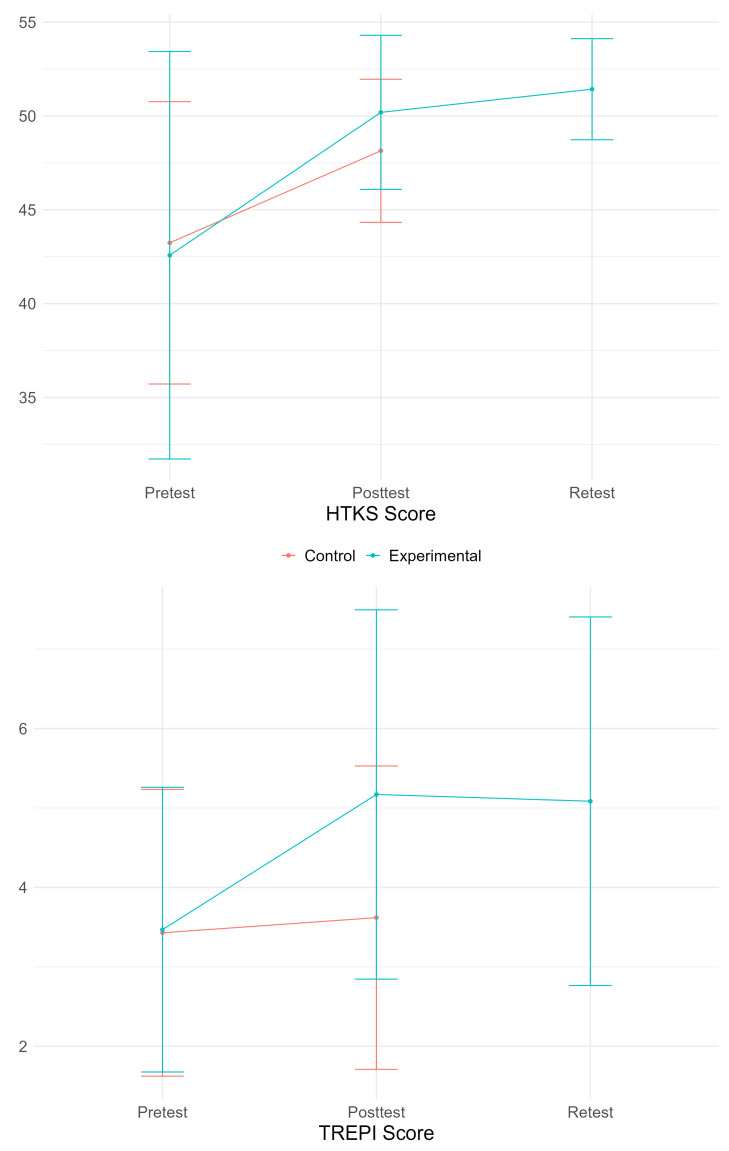
HTKS and TREPI scores in pretest, posttest, and retest measurements.

**Table 1 jfmk-09-00231-t001:** Study design and participants: Group I followed the experimental to control order and Group II control to experimental.

Group	Classroom	Sample	Pretest	Treatment	Posttest	Treatment	Retest
I	A and B	24 Boys 23 Girls	O1	ActivaMotricidad Program	O2	(No treatment)	O3
II	C	10 Boys 11 Girls	O1	(No treatment)	O2	ActivaMotricidad Program	--

**Table 3 jfmk-09-00231-t003:** Descriptive data aggregation of the study variables.

					95% CI	
Variable	Mean	SD	Minimum	Maximum	Lower Limit	Upper Limit
Age (months)	66.62	3.54	61	73	65.76	67.48
Active participation during recess	0.95	1.75	0.33	1	0.914	0.97
Participation during recess (seconds)	1710.00	185.87	720	1800.00	1665.00	1755.00
HTKS pre-experimental (Group I)	44.29	9.59	6	52	41.97	46.62
HTKS post-experimental (Group I)	50.71	3.49	34	52	49.86	51.55
HTKS pre-control (Group II)	48.04	6.24	24	52	46.53	49.55
HTKS post-control (Group II)	50.41	3.41	34	52	49.59	51.24
HTKS retest (only Group I)	51.43	2.69	34	52	50.63	52.22
TREPI pre-experimental (Group I)	3.51	1.81	0	9	3.08	3.95
TREPI post-experimental (Group I)	5.13	2.30	1	11	4.57	5.69
TREPI pre-control (Group II)	4.63	2.31	1	10	4.07	5.19
TREPI post-control (Group II)	4.63	2.29	0	10	4.08	5.19
TREPI retest (only Group I)	5.09	2.32	0	10	4.40	5.77

## Data Availability

The database and R code used in this study are freely available in the following online repository of the Open Science Framework: https://osf.io/cme6p/ (accessed on 9 October 2024).

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
