# Peer review of "Effect of the ActivaMotricidad Program on Improvements in Executive Functions and Interpersonal Relationships in Early Childhood Education"

_jfmk, 2024, doi:10.3390/jfmk9040231_

Round 1
Reviewer 1 Report
Comments and Suggestions for Authors
Basic reporting
Dear authors, the manuscript is generally well-written and easy to read; a slight spell-check is required. I have just some concerns that the authors must address.
Introduction
The literature on the subject is sufficiently well summarised. However, it could be useful to add some information about:
- you suggest a link between EF development, self-regulation, and physical activity, the causality is not always clear. For example, the claim that "fundamental motor skills training benefits cognitive development" implies a direct causal relationship without fully acknowledging potential confounding variables (e.g., socioeconomic factors, home environment, or genetic influences).
- you support the claim that play, especially cooperative motor games, enhances cognitive and social development, but it does not thoroughly address whether all types of play have the same effect. This implies that play in any form may lead to these improvements, which oversimplifies the complexity of play.
- you assume that all preschool children will benefit similarly from the "ActivaMotricidad" program, without addressing individual differences in learning styles, cognitive development, or social skills. This could be a methodological oversight when designing interventions.
Methods
- The sample size might be too small to detect small to moderate effects reliably, especially given the complexity of early childhood interventions that affect both cognitive and social domains. It would be helpful to include a power analysis to justify the sample size for detecting meaningful effects.
- Were all potential confounding variables (e.g., class size, teacher experience, socioeconomic factors) accounted for when randomizing?
- I could be wrong but, the control group had only one 50-minute Physical Education session per week, while the experimental group participated in various activities daily. This creates a significant disparity in the amount of physical activity between groups, which may confound the results.
- Did only the experimental group was subjected to a retention test, while the control group did not?
- Were the researchers implementing the program aware of the conditions?
- Did you have controlled diet, medication, or other lifestyle factors that could impact the results?
Validity of the findings
- The experimental group (47 children) is much larger than the control group (21 children). Unequal sample sizes between groups may introduce bias and limit the generalizability of the findings.
- The study implies that "active recess" contributed to positive outcomes but doesn’t account for potential confounding variables, such as the specific types of activities chosen by children, their interactions with peers, or differing play styles.
- There’s a lack of a discussion on potential biases from the teachers and children knowing they were part of an experimental program (i.e., the Hawthorne effect).
Comments on the Quality of English Languageminor issue detected
Author Response
Dear authors, the manuscript is generally well-written and easy to read; a slight spell-check is required. I have just some concerns that the authors must address.
We sincerely appreciate your review and the positive remarks regarding our manuscript. We have undertaken revisions to enhance the clarity of the text and to rectify minor errors. Below, we provide a detailed response to each of your suggestions, addressing them point by point, and changes in the manuscript are highlighted in yellow:
Introduction
The literature on the subject is sufficiently well summarised. However, it could be useful to add some information about:
- you suggest a link between EF development, self-regulation, and physical activity, the causality is not always clear. For example, the claim that "fundamental motor skills training benefits cognitive development" implies a direct causal relationship without fully acknowledging potential confounding variables (e.g., socioeconomic factors, home environment, or genetic influences).
We appreciate your observation regarding the need to clarify the causal relationship between the development of executive functions, self-regulation, and physical activity. In response, we have revised the Introduction section to address this concern and have included information on potential confounding factors that may influence the results (ll. 35-39).
- you support the claim that play, especially cooperative motor games, enhances cognitive and social development, but it does not thoroughly address whether all types of play have the same effect. This implies that play in any form may lead to these improvements, which oversimplifies the complexity of play.
You are correct. As noted in the work of D’Adamo & Lozada (2024), not all forms of play exert the same influence; additionally, complexity and context also play significant roles (Lyons, 2024). We have incorporated this information into the manuscript to clarify this point further (ll. 45-48).
- you assume that all preschool children will benefit similarly from the "ActivaMotricidad" program, without addressing individual differences in learning styles, cognitive development, or social skills. This could be a methodological oversight when designing interventions.
We appreciate your comment, as it addresses a point to which we paid considerable attention and attempted to explain, though it seems we did not do so adequately. In Table 2, under the section Cognitive Involvement, it is stated: “To increase the likelihood of reaching the children’s optimal challenge point, each task was designed with different incremental levels of difficulty (Howard et al., 2020; Schmidt et al., 2020): a) adding more elements to be remembered and/or manipulated in short-term storage (updating), b) introducing new rules requiring resolution of the change with the previous rule or a dominant behavior (inhibition), and c) including several sets of rules to switch fluidly (shifting).” Therefore, although there was no specific analysis of the individual capacity of the participants, there was monitoring to ensure that the tasks posed a challenge. We have made some changes to the manuscript to make this point clearer to the reader. First, in the limitations section, we have added a sentence with this idea (ll.426-432), and also in the methods section (l.206), a few lines in the program verification section. If you believe this point has not yet been adequately addressed, please let us know, as we want to make it very clear.
Methods
- The sample size might be too small to detect small to moderate effects reliably, especially given the complexity of early childhood interventions that affect both cognitive and social domains. It would be helpful to include a power analysis to justify the sample size for detecting meaningful effects.
Indeed, this has been a concern since the initial stages of the study design. Using the GPower software, we would need a sample of nearly 150 participants, which presents significant complexity and we believed that is not deemed necessary, as this study can be considered a preliminary approach to the effect of cognitive and motor activity with tasks whose difficulty is always adjusted, más aún en estas edades. Therefore, convinced of our ability to recruit the sample and that the results we would obtain would be sufficient to provide an understanding of the phenomena and that future studies could attempt to replicate with a larger sample, we opted to use non-parametric statistics. It is important to note that power analysis (such as using the GPower software) is traditionally applied to parametric tests. Non-parametric methods, such as the Mann-Whitney U test and the Wilcoxon test, are less sensitive to violations of normality assumptions and do not require the same sample size as parametric tests to be effective. These methods allow us to obtain robust and reliable results despite the limitations in sample size. In any case, you are absolutely correct, so we have included a few lines in this version in the Statistical Analysis section (ll. 306-311) and in the limitations (ll. 424-426) to clarify this point.
- Were all potential confounding variables (e.g., class size, teacher experience, socioeconomic factors) accounted for when randomizing?
We appreciate your highlighting this point, as we have realized that we mistakenly referred to our study design as solely a randomized controlled trial (RCT). In fact, it is a cluster RCT, as we randomized among the three classes available for the research, assigning two to the experimental group and one to the control group. We have incorporated these clarifications into the methodology section to provide a clearer understanding of the design. Consequently, as demonstrated in other studies that we have previously mentioned and referenced in the original manuscript, this type of design helps to control for certain confounding variables.
- I could be wrong but, the control group had only one 50-minute Physical Education session per week, while the experimental group participated in various activities daily. This creates a significant disparity in the amount of physical activity between groups, which may confound the results.
We appreciate the opportunity to clarify this point regarding our study, as we fully understand your concerns regarding the disparity in physical activity levels between the two groups. Firstly, as stated in the introduction, the objective of our research was to analyze the impact of increased volume of physical exercise with cognitive load during school hours, through the ActivaMotricidad program, on the executive functions and interpersonal skills of children aged 5 and 6 years. It is important to emphasize that our independent variable is physical exercise, not solely cognitive load.
We acknowledge that the difference in physical activity may influence the results. However, previous studies, such as that by Pesce (2021), have demonstrated that cognitive load during exercise significantly impacts executive functions, even in contexts where the control group maintains its usual routine. In this regard, the design of our study aligns with common practices in ecological research, where the control group engages in their habitual activities while the experimental group participates in a more intensive program.
Examples of this can be found in other articles we have cited in this manuscript:
- Robinson et al. (2016): In their study, the experimental group replaced their recess with 40 minutes of exercise, without controlling the activity level of the control group during recess.
- Özkür et al. (2019): They proposed an integrative program for the experimental group, while the control group maintained its usual routine.
- Howard et al. (2020): The control group continued with its existing program, which included structured and free playtime.
Any of the characteristics of the ActivaMotricidad program could be responsible for the observed improvements. While one might hypothesize that merely increasing the volume of physical exercise is sufficient to yield beneficial results, previous studies do not support this explanation, as we have indicated in the introduction. Our intention is to integrate all variables that have been found to positively influence executive functions in childhood and to consolidate them into a feasible and realistic intervention program.
- Did only the experimental group was subjected to a retention test, while the control group did not?
Thanks for the opportunity to clarify this point because it is important. As indicated in the design section of the manuscript (ll. 148-149), the control group received a delayed implementation of the program after the trial concluded, allowing them to benefit from its potential positive effects. This decision was made because we believe that if the intervention is hypothesized to have a positive effect on individuals, it is ethically imperative for all participants in the study to receive the intervention.
Given our constraints regarding time, we could not wait to conduct the retention measurement for the control group before administering the intervention. Consequently, we were unable to perform the retention measurement for the control group. Since the control group did not undergo this measurement, we believe that this strategy is valid and contributes to understanding the effects of the program in both groups.
If the reviewer feels that additional information should be included in the manuscript to enhance clarity on this matter, we would be happy to accommodate that request.
- Were the researchers implementing the program aware of the conditions?
In the manuscript, we mention in the design section that the teachers were unaware of the study conditions and were informed that they would receive access to the program resources once the study was completed (ll.145-146). We believe this information is clear; however, we are open to considering any adjustments that may enhance the clarity of the text.
- Did you have controlled diet, medication, or other lifestyle factors that could impact the results?
We appreciate your question, which is entirely understandable given that the manuscript may have led to the interpretation of the design as a randomized controlled trial (RCT). In reality, it is a cluster RCT, where we assigned two classes to the experimental group and one class to the control group. This approach allows for better control of the external factors you mentioned; however, we acknowledge that specifically measuring these factors could enhance the study. Consequently, we have included this information in the limitations section (ll. 432-434).
Validity of the findings
- The experimental group (47 children) is much larger than the control group (21 children). Unequal sample sizes between groups may introduce bias and limit the generalizability of the findings.
We acknowledge that the experimental group is larger than the control group, which could introduce biases and limit the generalizability of the findings. However, as previously mentioned, we have considered this issue from the outset. Therefore, we opted to employ non-parametric statistics to obtain robust and reliable results. Nonetheless, we recognize that future studies should be conducted with a larger sample size and different statistical methods.
- The study implies that "active recess" contributed to positive outcomes but doesn’t account for potential confounding variables, such as the specific types of activities chosen by children, their interactions with peers, or differing play styles.
We appreciate your comment regarding confounding variables. In this article, to determine whether the sample was physically active during active recess, we utilized the Active Recess Observation System in Early Childhood (SOREACI) to assess the children’s motor engagement during unstructured playtime. This system allowed us to categorize the children based on their physical activity levels and to directly observe their behavior in the playground environment, from which we found that they were physically active and homogeneous across the sample.
While SOREACI provides a robust framework for observation and we consider it valid for our research purposes, we acknowledge that factors such as the specific types of activities chosen by the children, their interactions with peers, and their play styles represent limitations that must be considered. We have included this information in the limitations section (ll. 434-437).
- There’s a lack of a discussion on potential biases from the teachers and children knowing they were part of an experimental program (i.e., the Hawthorne effect).
We find it truly interesting to discuss this matter, and we appreciate the opportunity for dialogue. Firstly, we acknowledge that this phenomenon can occur in any research and may influence participants’ behavior when they know they are being observed. However, we consider this effect to be a common aspect in most studies.
In our research, the hypotheses are fulfilled, and the significant results cannot be attributed solely to the potential Hawthorne effect. Our objective is to integrate all variables that have been shown to positively influence executive functions in childhood into a practical and applicable intervention program. Notably, other studies have reported similar findings without mentioning this effect.
Therefore, while we are open to including a discussion on the Hawthorne effect if deemed necessary, we believe it is not essential for the clarity of our study, as it may confuse the reader and unnecessarily extend the text.
Reviewer 2 Report
Comments and Suggestions for Authors
Thanks for inviting me to review this interesting study. I have two minor concerns.
First is about the researd design: The authors claimed that they conducted Randomized Controlled Trial (RCT) in this study. In my understanding, RCT is individualized. That means, participants are randomly assigned to either the intervention or control group. Its focus is to evaluates the effect of an intervention on individual outcomes. Instead, this study conducted a cluster Randomized Controlled Trial (Cluster RCT). This means a class of participants is randomly assigned to intervention or control groups. This Cluster RCT aims to evaluate the effect of an intervention on group-level outcomes and is useful when individual randomization is impractical or when the intervention is designed to be delivered at the group level.
The second is about program fidelity, which refers to the degree to which an intervention or program is delivered as intended by its developers. It's about ensuring the original design is faithfully implemented to maintain its effectiveness. It includes: (1) Adherence to the core components; (2)
Quality of delivery; (3) Participant responsiveness.
I would appreciate it if the authors could improve their manuscript by addressing the two minors. Thanks!
Comments on the Quality of English LanguageFine with me. Better to have it edited when R1 is completed.
Author Response
Thanks for inviting me to review this interesting study. I have two minor concerns.
We sincerely appreciate your review and the time you have dedicated to it. Below, we address your two minor concerns point by point, and changes in the manuscript are highlighted in yellow.
First is about the researd design: The authors claimed that they conducted Randomized Controlled Trial (RCT) in this study. In my understanding, RCT is individualized. That means, participants are randomly assigned to either the intervention or control group. Its focus is to evaluates the effect of an intervention on individual outcomes. Instead, this study conducted a cluster Randomized Controlled Trial (Cluster RCT). This means a class of participants is randomly assigned to intervention or control groups. This Cluster RCT aims to evaluate the effect of an intervention on group-level outcomes and is useful when individual randomization is impractical or when the intervention is designed to be delivered at the group level.
You are absolutely correct, and we appreciate you bringing this to our attention. Indeed, the design used was a cluster RCT. In the manuscript, we stated that the randomization of the study conditions was done between classes, with two classes randomly assigned as the experimental condition and one as the control condition. Additionally, in the methodology, we mentioned that other studies followed this design as valid, also randomizing by classes.
However, it is true that we did not explicitly use the term “cluster,” which could lead to confusion for the reader. This issue can be easily resolved thanks to the reviewer’s comment. Therefore, following your suggestions, we have changed all mentions of RCT to cluster RCT throughout the manuscript to avoid any confusion. Thank you very much for pointing this out.
The second is about program fidelity, which refers to the degree to which an intervention or program is delivered as intended by its developers. It's about ensuring the original design is faithfully implemented to maintain its effectiveness. It includes: (1) Adherence to the core components; (2) Quality of delivery; (3) Participant responsiveness.
We sincerely appreciate your comment regarding program fidelity, which is a crucial aspect of any intervention’s effectiveness. We want to assure you that we have considered this topic from the beginning of our study. We believe that the original manuscript included several elements addressing the components of program fidelity. However, this information was not presented clearly, and we also lacked some relevant details. We have now added this information (see ll. 226) and have reorganized certain sections of the methods to make this point easier to identify.
Therefore, we believe that, as how it is presented now in the article, every part of the program fidelity can be inferred and was effectively implemented. However, if the reviewer suggests that we should present this information in a specific format, we are open to your feedback and willing to include it in the next version.
Reviewer 3 Report
Comments and Suggestions for Authors
the study presents a very good design and shows interesting data;
the authors state that the study did not have a control group, put that information in the paragraph of the study limitation;
in table 2, authors refer in terms of cognitive impairment:
To increase the likelihood of reaching the children’s optimal challenge point, each task was designed with different incremental levels of difficulty (Howard et al., 2020; Schmidt et al., 2020). (what were this tasks? please describe...
a) adding more elements to be remembered and/or manipulated in short-term storage (updating); which elements?? please describe...
b) introducing new rules requiring resolution of the change with the previous rule or a dominant behavior (inhibition); which rules?? please describe...
there are many references citations with two differente types (ones with numbers and others have the authors names cited in the text. must be standardized. i think the journal requirements are numbers according to the authors citations in the text.
Author Response
The study presents a very good design and shows interesting data;
We appreciate the effort you have put into reviewing our article and your valuable suggestions. Below, we address each of your comments point by point, and changes in the manuscript are highlighted in yellow.
The authors state that the study did not have a control group, put that information in the paragraph of the study limitation;
We have made an effort to verify that it was not stated that there is no control group, as this study does indeed include a control group (comprising one class) and an experimental group (comprising two classes). We have reviewed the abstract, objectives, methodology, and results, and we did not find any indication suggesting the absence of a control group.
However, it is possible that this confusion arose from the wording in certain sections of the methodology. We have made revisions to clarify that a control group was indeed present and to explain how it was configured (ll. 139-142; 156).
In table 2, authors refer in terms of cognitive impairment:
To increase the likelihood of reaching the children’s optimal challenge point, each task was designed with different incremental levels of difficulty (Howard et al., 2020; Schmidt et al., 2020). (what were this tasks? please describe...
- a) adding more elements to be remembered and/or manipulated in short-term storage (updating); which elements?? please describe...
- b) introducing new rules requiring resolution of the change with the previous rule or a dominant behavior (inhibition); which rules?? please describe...
Thank you for your comments and questions regarding the tasks designed to enhance the optimal challenge for children. The description of the elements you mentioned was indeed included in an open link associated with Figure 1. While some of this information was present in the methodology, it was not detailed extensively to avoid lengthening the text unnecessarily.
Below, I have provided the specific information you requested:
Designed Tasks
- Sweet and Savory Word Task:
- Description: Each child is assigned a word (either “sweet” or “savory”) and must remember their assignment while walking. When the teacher says one of the words, only the children with that word should jump.
- Elements to update:
- Elements to remember: Children need to remember their assigned word, and in variants, they may remember two words (e.g., “sweet” and “chocolate”).
- Rules to inhibit:
- Rules: In the reverse play variant, if the teacher says “savory,” children with “sweet” must jump, requiring them to inhibit their usual response.
- Calm and Attentive Like a Frog Activity:
- Description: Children practice deep breathing and attention, mimicking the behavior of a frog.
- Elements to update:
- Elements to remember: Children should recall five things they saw on their way to school, which involves a short-term memory exercise.
- Rules to inhibit:
- Rules: During the activity, they must remain still when the teacher says “calm” and move only when instructed “wind,” requiring inhibition of spontaneous movement.
- The Bandit Wind Narration:
- Description: Children act as the zebra Camila, moving freely until instructed to freeze.
- Elements to update:
- Elements to remember: They must remember the instructions on when to move and when to freeze.
- Rules to inhibit:
- Rules: They must find their group and hug each other when the teacher says “wind,” which requires inhibiting individual movement to collaborate as a group.
In any case, to clarify that readers who wish to know specifically how these tasks were structured must access the open link, we have rewritten the reference to Figure 1 for greater clarity (ll. 200-204). If the reviewer believes that this information should be included in the manuscript, or perhaps in an appendix, we are open to making that adjustment.
There are many references citations with two differente types (ones with numbers and others have the authors names cited in the text. must be standardized. i think the journal requirements are numbers according to the authors citations in the text.
We apologize for any confusion. In fact, we have not employed two different styles; we have consistently followed the journal’s guidelines, which specify the use of numbers for citations. However, in instances where we wanted to emphasize the authors’ names, we included the number instead of the year, as illustrated in the following example:
“Previous studies have demonstrated the effects of cooperative games on social, emotional, and psychomotor development variables [13]. Recently, D’Adamo & Lozada [14] found that children’s participation in playful and active interventions contributed to improving their social relationships. Similarly, Lyons [15] observed that the inclusion of play within the classroom in early childhood education had a positive effect on interpersonal skills after a four-month period.”
We understand that this may create the impression of using two distinct styles. Therefore, we have made revisions throughout the manuscript to standardize the reference citations (ll. 41-43; 59-60; 185-186; 371-372; 380-383; 403-404; 414-415).
Round 2
Reviewer 1 Report
Comments and Suggestions for Authors
the authors adressed all my concerns. I have no further suggestions.